# In Vitro Cytotoxic Potential and In Vivo Antitumor Effects of NOS/PDK-Inhibitor T1084

**DOI:** 10.3390/ijms25179711

**Published:** 2024-09-08

**Authors:** Marina Filimonova, Anna Shitova, Ljudmila Shevchenko, Olga Soldatova, Valentina Surinova, Vitaly Rybachuk, Alexander Kosachenko, Kirill Nikolaev, Irina Volkova, Grigory Demyashkin, Tatjana P. Stanojkovic, Zeljko Zizak, Sergey Ivanov, Petr Shegay, Andrey Kaprin, Alexander Filimonov

**Affiliations:** 1A. Tsyb Medical Radiological Research Center—Branch of the National Medical Research Radiological Center of the Ministry of Health of the Russian Federation, 249036 Obninsk, Russia; annaredrose@mail.ru (A.S.); schew.ludmila@yandex.ru (L.S.); 89208861291@mail.ru (O.S.); val_suriniva@mail.ru (V.S.); rybachukvitaliy@gmail.com (V.R.); br.shepard@list.ru (A.K.); nireallki@gmail.com (K.N.); ik_volkova@mail.ru (I.V.); dr.dga@mail.ru (G.D.); filimonov_alex@mail.ru (A.F.); 2Department of Experimental Oncology, Laboratory for Radiobiology and Experimental Oncology, Institute of Oncology and Radiology of Serbia, Pasterova 14, 11000 Belgrade, Serbia; 3National Medical Research Radiological Center of the Ministry of Health of the Russian Federation, 249036 Obninsk, Russia; dr.shegai@mail.ru (P.S.); kaprin@mail.ru (A.K.)

**Keywords:** NOS-inhibitor, PDK-inhibitor, cytotoxic effect, antitumor effect, hypoxic resistance, cervical cancer, melanoma

## Abstract

Previously, we showed the antitumor activity of the new NOS/PDK inhibitor T1084 (1-isobutanoyl-2-isopropylisothiourea dichloroacetate). The present study included an assessment of in vitro cytotoxicity against human malignant and normal cells according to the MTT-test and in vivo antitumor effects in solid tumor models in comparison with precursor compounds T1023 (NOS inhibitor; 1-isobutanoyl-2-isopropylisothiourea hydrobromide) and Na-DCA (PDK inhibitor; sodium dichloroacetate), using morphological, histological, and immunohistochemical methods. The effects of T1084 and T1023 on the in vitro survival of normal (MRC-5) and most malignant cells (A375, MFC-7, K562, OAW42, and PC-3) were similar and quantitatively equal. At the same time, melanoma A375 cells showed 2–2.5 times higher sensitivity (IC_50_: 0.39–0.41 mM) to the cytotoxicity of T1023 and T1084 than other cells. And only HeLa cells showed significantly higher sensitivity to the cytotoxicity of T1084 compared to T1023 (IC_50_: 0.54 ± 0.03 and 0.81 ± 0.02 mM). Comparative studies of the in vivo antitumor effects of Na-DCA, T1023, and T1084 on CC-5 cervical cancer and B-16 melanoma in mice were conducted with subchronic daily i.p. administration of these agents at an equimolar dose of 0.22 mmol/kg (33.6, 60.0, and 70.7 mg/kg, respectively). Cervical cancer CC-5 fairly quickly evaded the effects of both Na-DCA and T1023. So, from the end of the first week of Na-DCA or T1023 treatment, the tumor growth inhibition (TGI) began to decrease from 40% to an insignificant level by the end of the observation. In contrast, in two independent experiments, CC-5 showed consistently high sensitivity to the action of T1084: a significant antitumor effect with high TGI (43–58%) was registered throughout the observation, without any signs of neoplasia adaptation. The effect of precursor compounds on melanoma B-16 was either minimal (for Na-DCA) or moderate (for T1023) with TGI only 33%, which subsequently decreased by the end of the experiment. In contrast, the effect of T1084 on B-16 was qualitatively more pronounced and steadily increasing; it was accompanied by a 3-fold expansion of necrosis and dystrophy areas, a decrease in proliferation, and increased apoptosis of tumor cells. Morphologically, the T1084 effect was 2-fold superior to the effects of T1023—the TGI index reached 59–62%. This study suggests that the antitumor effects of T1084 develop through the interaction of NOS-dependent and PDK-dependent pathophysiological effects of this NOS/PDK inhibitor. The NOS inhibitory activity of T1084 exerts an anti-angiogenic effect on neoplasia. At the same time, the PDK inhibitory activity of T1084 enhances the cytotoxicity of induced intratumoral hypoxia and suppresses the development of neoplasia adaptation to anti-angiogenic stress. Such properties allow T1084 to overcome tumor resistance and realize a stable synergistic antitumor effect.

## 1. Introduction

More than half a century ago, in 1971, Dr. Judah Folkman substantiated the role of angiogenesis in tumor progression [1]. Since then, anti-angiogenic therapy has been considered as one of the promising treatment strategies for solid cancers. The first anti-angiogenic drug, bevacizumab, was first approved for the treatment of metastatic colorectal cancer in 2004. And this was perceived as a big step forward in cancer treatment. To date, the range of anti-angiogenic targets and drugs has expanded significantly, and their use in clinical oncology has become significant. However, despite these successes, initial optimistic expectations have not yet been achieved, and the clinical effectiveness of anti-angiogenic therapy remains moderate [2,3,4,5]. Indeed, available anti-angiogenic agents are not effective against all solid cancers. As a monotherapy, the inhibition of tumor angiogenesis is able to arrest or slow tumor growth, but cannot eliminate the tumor. When combined with other anticancer treatment methods, anti-angiogenic therapy is capable of improving survival (by several months) for most cancer patients. But in some patients, due to intrinsic refractoriness or rapid development of resistance, anti-angiogenic therapy has little or no beneficial effect. The primary or acquired resistance towards anti-angiogenic drugs still remains a common problem in cancer treatment. Obviously, to change the current picture, the further detailed study of the pathophysiology of tumor angiogenesis and the mechanisms of tumor resistance, the search for new molecular targets, and the development of new strategies for anti-angiogenic cancer therapy are necessary [3,4,6,7,8].

A lot of data that indicate the active multifaceted participation of endogenous nitric oxide (NO) in angiogenesis and tumor progression have been obtained in recent decades [9,10,11,12,13,14,15]. In this regard, the ability of nitric oxide synthase (NOS) inhibitors to exert an anti-angiogenic effect on solid tumors [16,17,18,19,20,21,22] seems quite expected. One of them, 1-isobutanoyl-2-isopropylisothiourea hydrobromide (compound T1023; Figure 1A), is an effective competitive inhibitor of iNOS/eNOS (IC_50_ for nNOS, iNOS, eNOS: 52.3, 3.2, and 5.1 μM, respectively). Its chronic parenteral administration at safe doses of 30–60 mg/kg (1/9–1/5 LD_16_) causes the inhibition of tumor growth and suppression of metastasis in a number of transplanted solid tumors in mice, and synergistically increases the antitumor effect of bevacizumab, cyclophosphamide, and γ-rays [23,24,25,26].

These effects of T1023 were realized primarily through the anti-angiogenic method. So, a pronounced (1.4–1.6 times) decrease in the content of blood vessels in the peritumoral area and “hot spots” of angiogenesis in the parenchyma of tumor nodes and an increase (1.5–2 times) in the fraction of hypoxic (pO_2_ < 10 mmHg) tumor cells were observed on the third–fourth day from the start of T1023 administration. Such disturbances in tumor oxygenation during this period were accompanied by a decrease in proliferation (1.3–1.6 times) and a sharp increase (2–3 times) in the apoptotic death of tumor cells in the tumor parenchyma [24,25].

Nevertheless, the antitumor efficacy of T1023 with chronic use, as a rule, decreased in the long term of development of various experimental tumors (Lewis lung carcinoma, Ehrlich carcinoma, melanoma B-16). During the long-term period of T1023 therapy, deep hypoxia and pronounced disturbances in tumor vascularization remained in the tumor nodes. But at the same time, apoptotic death decreased, whereas the proliferative activity of tumor cells increased, and as a consequence, recurrent tumor growth developed in T1023-treated mice. Apparently, this was a manifestation of the development of resistance (adaptation) to experimental neoplasia to the anti-angiogenic effect of T1023.

Tumor resistance to anti-angiogenic therapy is obviously provided by multiple mechanisms that may be activated by genetic, epigenetic, and microenvironmental factors [3,4,6,7,8]. However, it can be assumed that the cellular substrate for tumor adaptation to anti-angiogenic stress is hypoxic tumor cells and the microenvironment. Based on these positions, the way to increase antitumor effectiveness is through the use of T1023 in combination with hypoxic toxins, which mainly affect hypoxic cells. 

Currently, a significant number of hypoxia-oriented targets and chemical agents acting on them are known [27,28,29]. One of the hypoxia-targeted cytotoxins is pyruvate dehydrogenase kinase (PDK) inhibitor—dichloroacetate (DCA; Figure 1B). DCA is able to induce apoptosis of hypoxic cells by stimulating the activity of the Krebs cycle and the mitochondrial respiratory chain [30,31]. The sensitivity of many tumors to the DCA action has been demonstrated. And a number of studies have noted the ability of DCA to overcome resistance to anticancer and anti-angiogenic drugs [32,33,34,35,36].

Initially, we assessed the promise of the combination of T1023 and DCA (in the form of sodium salt—Na-DCA) on the Ehrlich solid carcinoma model [37,38]. It was found that the combined use of T1023 (daily, i.p., 60 mg/kg) and Na-DCA (every other day, i.g., 162 mg/kg) causes a synergistic antitumor effect in comparison with the separate use of these agents. However, a week after the start of monotherapy with T1023 or Na-DCA, an increase in tumor resistance to this treatment was observed, and in the later stages of the experiment, the antitumor effects of monotherapy were minimal. At the same time, no increase in tumor resistance to the combination of T1023 and Na-DCA was observed. Mice receiving the combination treatment demonstrated a pronounced and stable antitumor effect throughout the observation period.

These results inspired us to develop another approach to the combined use of T1023 and DCA. The NOS-inhibiting fragment of T1023, 1-isobutanoyl-2-isopropylisothiourea, is a weak base, and DCA is a fairly strong organic acid. This made it possible to chemically construct a compound combining NOS-inhibiting and PDK-inhibiting fragments in the molecular structure—1-isobutanoyl-2-isopropylisothiourea dichloroacetate salt (compound T1084; Figure 1C). In solution, most of the molecules of this salt will dissociate into ions, so the new compound T1084 could potentially be capable of implementing both types of biochemical activity: NOS inhibition due to the cation ion and PDK inhibition due to the anion ion.

We managed to develop a method and synthetically realize this compound [39]. The results of the first study of T1084 [40], in general, confirmed the promise of the proposed approach to the development of multifunctional antitumor agents. Biochemical methods confirmed that T1084 is indeed a bifunctional NOS/PDK inhibitor. The new compound T1084 realizes in vivo NOS-inhibiting and PDK-inhibiting activity, quantitatively, at the level of the previous compounds, T1023 and Na-DCA, at equimolar doses. A pronounced synergistic antitumor effect of T1084 in comparison with T1023 and Na-DCA at equimolar doses was observed in two independent experiments on the Ehrlich solid carcinoma model. There were no signs of tumor adaptation to T1084 treatment, while experimental neoplasia rapidly desensitized to the separate treatment of both T1023 and Na-DCA.

In continuation of the antitumor efficacy studies of the NOS/PDK inhibitor T1084, we assessed its in vitro cytotoxic potential as well as its in vivo antitumor effects in solid tumor models using histological, immunohistochemical, and morphometric methods in comparison with the precursor compounds T1023 and DCA.

## 2. Results

### 2.1. In Vitro Cytotoxic Activity of T1023 and T1084

The compounds T1023 and T1084 were tested for in vitro cytotoxic activity against six human malignant cell lines (A375, MFC-7, K562, HeLa, OAW42, and PC-3) and one normal human cell line (MRC-5). The results are presented in Figure 2 in the form of representative cell survival curves, the IC_50_ values, and the selectivity index values, calculated as the ratios between the IC_50_ values obtained on MRC-5 and malignant cells.

The data obtained indicated a similar, statistically equal cytotoxic effect of both T1023 and T1084 on the majority of cell lines (A375, MFC-7, K562, OAW42, PC-3, and MRC-5). At the same time, the cytotoxicity of T1023 and 1084 against most of these cell lines (MFC-7, K562, OAW42, PC-3, and MRC-5) was nonspecific. The IC_50_ values obtained on these cell lines (0.65–1.41 mM) corresponded to subtoxic and toxic doses (more than 0.7×LD_16_) of these compounds in the in vivo acute lethality test [40]. The exception in this series was the human melanoma A375 cell line. A375 cells showed 2–2.5 times higher sensitivity to the cytotoxic effects of T1023 and T1084 than other cell lines studied. In this case, the IC_50_ values (0.39–0.41 mM) were at a safer toxicological level (about 0.4×LD_16_), and the high values of the A375 cells’ selectivity index (2.2–2.4) suggest the presence of the specific cytotoxicity of T1023 and T1084 to melanoma. This study also brought attention to the human cervical cancer HeLa cell line. HeLa cells showed low sensitivity to the cytotoxicity of T1023 (IC_50_—0.81 ± 0.02 mM), but the cytotoxic effect of T1084 against these cells was significantly higher (IC_50_—0.54 ± 0.03 mM). The high value of the T1084 selectivity index for HeLa cells (1.7) also suggests the presence of the specific cytotoxicity of T1084 against cervical cancer.

These results justified the feasibility of studying the in vivo antitumor effects of T1084 in comparison with its precursor compounds, Na-DCA and T1023, on two solid tumor models in mice: cervical cancer CC-5 and B-16 melanoma.

### 2.2. In Vivo Antitumor Effect of Na-DCA, T1023 and T1084 on Cervical Cancer CC-5

Two independent experiments were performed on this tumor model. The first experiment on cervical cancer CC-5 was carried out on 80 female CBA mice, divided into a control and three experimental groups (20 mice in each). Experimental treatment began on the seventh day after tumor inoculation. From this day until the 20th day of tumor growth, the mice of the experimental groups were injected i.p. daily with T1023, Na-DCA, or T1084 at equimolar doses of 0.22 mmol/kg (60.0, 33.6, and 70.7 mg/kg, respectively). The mice of the control groups were injected i.p. daily with 0.9% sodium chloride in equivalent volume. The duration of the experiments was 23 days from the tumor transplantation.

The results of the first experiment showed (Figure 3A,C) that initially this neoplasia was highly sensitive to the action of all tested compounds. Already after two injections of Na-DCA, T1023, or T1084 in equimolar doses, significant antitumor effects were observed in experimental groups of mice, accompanied by significant inhibition of tumor growth (by 40% when exposed to Na-DCA and T1023; by 55% when exposed to T1084). But the further dynamics of the antitumor effects in the groups differed qualitatively. 

Further administration of Na-DCA as well as T1023 was accompanied by a rapid decrease in the sensitivity of CC-5 to the action of both agents. Already from the fifth day of treatment until the end of observation, the antitumor effects and inhibition of tumor growth had reached an insignificant level in the mice of these groups. The integral growth retardation of CC-5 induced by both Na-DCA and T1023 in this experiment was weak—the time for a 10-fold increase in tumor volume increased by only 1.1 days (+16%) in these groups. At the same time, no significant morphological manifestations of CC-5 adaptation to the ongoing administration of T1084 were observed. In T1084-treated mice, a highly significant antitumor effect was observed throughout the experiment, accompanied by stable inhibition (by 48–58%) of tumor growth. The integral delay in CC-5 growth induced by T1084 was also significant—the time for a 10-fold increase in tumor volume in this group increased by 4.3 days (+64%). The significance of the qualitative differences in the antitumor effects of T1084 and its precursor compounds, Na-DCA and T1023, was confirmed by high levels of intergroup statistical differences in tumor volume and tumor growth inhibition during the observation stages (Figure 3A,C).

The second experiment on cervical cancer CC-5 was carried out on 40 female CBA mice, divided into control and experimental groups (20 mice in each). Daily i.p. administration of 0.9% sodium chloride or T1084 at dose of 0.22 mmol/kg (70.7 mg/kg) was carried out from the eighth to twentieth days of tumor growth. The total duration of observation was 24 days after tumor transplantation.

The results of this experiment confirmed the ability of T1084 to have a pronounced and stable antitumor effect against CC-5 and the absence of the significant adaptation of neoplasia to NOS/PDK inhibitor subchronic exposure (Figure 3B,D). In T1084-treated mice, as in the first study, a highly significant antitumor effect was observed throughout the experiment, accompanied by stable inhibition (43–52%) of tumor growth. The integral delay in CC-5 growth induced by T1084 was also significant—the time for a 10-fold increase in tumor volume in this group increased by 4.0 days (+56%). Moreover, in this experiment, a significant duration of T1084 action was also observed; no noticeable weakening of its effects was registered within 5 days after the completion of treatment.

### 2.3. In Vivo Antitumor Effect of Na-DCA, T1023, and T1084 on Melanoma B-16

The experiment on B-16 melanoma was carried out on 88 female C57Bl/6j mice, divided into a control and three experimental groups (22 mice in each). Experimental treatment began on the eighth day after tumor inoculation. From this day until the 20th day of tumor growth, the mice of the experimental groups were injected i.p. daily with T1023, Na-DCA, or T1084 at equimolar doses of 0.22 mmol/kg (60.0, 33.6, and 70.7 mg/kg, respectively). The mice of the control groups were injected i.p. daily with 0.9% sodium chloride in equivalent volume. The duration of the experiments was 21 days from the tumor transplantation. In this experiment, on the thirteenth day of melanoma growth (sixth day from the beginning of the exposure), seven mice from each group were bred for pathomorphological and histological studies.

The morphological results of this experiment showed (Figure 4) that initially this neoplasia had different sensitivity to the action of the studied compounds. Thus, melanoma B-16 was consistently insensitive to Na-DCA subchronic exposure. During the entire experiment, no significant antitumor effects were observed in Na-DCA-treated mice; tumor growth inhibition in this group did not exceed 20%, and melanoma B-16 tumor nodes in these mice at the final stage did not differ from the control in appearance or tumor weight.

However, melanoma B-16 was moderately sensitive to subchronic exposure to T1023. The antitumor effect of T1023 gradually increased over the course of treatment. A significant effect of T1023 developed after five injections and reached a maximum (tumor growth inhibition of 30–33%) after seven–nine injections. But at the later stages of the experiment, a tendency for neoplasia to adapt to T1023 action was registered; the effect was noticeably weakened (tumor growth inhibition: 27%). Overall, the integral growth delay of B-16 induced by T1023 was moderate—the time to a 6-fold increase in tumor volume in this group increased by 2.3 days (+37%). However, melanoma B-16 tumor nodes in T1023-treated mice at the final stage were smaller in appearance and tumor weight than control tumors.

At the same time, melanoma B-16 was consistently highly sensitive to subchronic exposure to T1084. The antitumor effect developed rapidly and increased throughout the course of treatment. A significant effect of T1084 was observed after two injections and reached a maximum (tumor growth inhibition—59–62%) after nine–thirteen injections. No manifestations of neoplasia adaptation to the action of T1084 were observed. The integral growth delay of B-16 induced by T1084 was also high—the time for a 6-fold increase in tumor volume in this group increased by 6.4 days (+102%). The tumor nodes of melanoma B-16 in T1024-treated mice at the final stage were smaller in appearance and tumor weight than in T1023-treated mice. The significance of the qualitative differences in the antitumor effects of T1084 and its precursor compounds, Na-DCA and T1023, was confirmed by high levels of intergroup statistical differences in tumor volume and tumor growth inhibition at the follow-up stages (Figure 4).

In this experiment, comparative pathomorphological, histological, and immunohistochemical studies of B-16 melanoma tumor nodes at the exponential stage of growth (thirteenth day after transplantation; sixth day after the start of exposure) were also carried out in control mice and animals of the experimental groups that received Na-DCA, T1023, or T1084 injections.

In the mice of all groups, the malignant neoplasm was located (implanted) under thin skin and, in some places, surrounded by muscle fibers with mild degenerative changes. Tumor: melanoma (partially pigmented) nodular type (Figure 5A). 

In the mice of the control group, most of the parenchyma of B16 melanoma is represented by fields of polymorphic cells filling the tumor nodes. In areas of solid structure located in the subcutaneous tissue and at the border with the underlying muscles, tumor cells are densely packed. In interphase cells, the nuclei are large with fine-grained chromatin and large nucleoli (Figure 5B,C). In the fields of view, relatively numerous mitotic figures and single cells dying by apoptosis are observed. In the proximal part of the tumor nodes, a pronounced growth of melanoma is observed, with strands of tumor cells spreading between the fibers of the striated muscles.

The peritumoral zone of the tumors consists of a narrow layer of loose, unformed connective tissue containing small and larger blood vessels. Some larger vessels are dilated, and accumulations of tumor cells are found in their lumen (vascular invasion). In the subcutaneous tissue adjacent to the tumor, there is intense infiltration of round cell elements (cellular inflammatory infiltration), among which lymphocytes predominate.

In the central and peripheral parts of the tumor nodes there are narrow oxyphilic layers of spontaneous necrosis, extended in the proximodistal direction and represented by cellular detritus, with perinecrotic rims of atypical Ki-67-immunonegative cells in a state of dystrophy (Table 1).

In depigmented sections of B-16 melanoma from control animals, almost all blood vessels showed an intense positive endothelial reaction for CD31 both at the periphery of the tumor and in “hot spots” of its parenchyma (Figure 6; Table 2). The majority of atypical cell nuclei (about 82%) in sections of the control tumors gave positive immunohistochemical staining for Ki-67 (Figure 7; Table 3). Positively stained nuclei for Caspase-3 in these tumors, on the contrary, were detected in low numbers (10–14 per mm^2^) (Figure 8; Table 4).

A study of the review preparations of tumors in Na-DCA-treated mice did not reveal significant changes in the histological pattern of B16 melanoma, compared with the control (Figure 5A). However, attention was drawn to the features in the peripheral areas of the tumors, especially from the subcutaneous tissue: an increase in the content of narrow linear layers of necrosis, surrounded by belts of dystrophically changed atypical cells (Figure 5B,C; Table 1). In addition, a decrease in the intensity of the Ki-67-positive reaction of tumor cell nuclei and an increase in the proportion of Caspase-3-positive atypical cells was observed in these areas (Figure 7 and Figure 8). These changes were noted against the background of increased polymorphism of the melanoma angioarchitecture. In the proximal zone of the tumor nodes, the blood vessels are dilated, their endothelium is not visible in places, and the surrounding stroma looked edematous. Plasmorrhagia and microhemorrhages are sometimes detected. In areas of the solid structure of the tumor parenchyma, most hemocapillaries are slit-shaped. In the H&E-stained sections, vascular endothelial nuclei appear flattened and hyperchromatic, and some of them are pyknotic.

In depigmented melanoma sections of Na-DCA-treated mice, almost all blood vessels showed a positive endothelial reaction to CD31 both in the periphery of the tumor and in its parenchyma (Figure 6; Table 2). The vascular network in the tumor parenchyma is unevenly distributed. In the immunohistochemical reaction to Ki-67, the number of positively stained nuclei of proliferating atypical cells is moderately but statistically significantly reduced (up to 74%) compared to the control (Figure 7; Table 3). Positively stained nuclei for Caspase-3 in these tumors, on the contrary, are detected in slightly greater numbers (15–23 per mm^2^) than in the control tumors (Figure 8; Table 4).

In the review preparations of tumors from T1023-treated mice, the melanoma parenchyma also appeared relatively intact (Figure 5A). But the presence of significant pathological changes in the peritumoral vascular network attracted attention. Visually, the content of vessels in the subcutaneous tissue is reduced due to a decrease in the number of narrow, unevenly dilated vessels of the sinusoidal type, both at the border of tumor and healthy tissue, and in areas of solid parenchyma structure. In microslides stained with H&E, in some places only eosinophilic contours of dead microvessels are identified (Figure 5B,C). Against the background of such pronounced vascular disorders, branched islands of necrosis with perinecrotic rims of cells in a state of dystrophy are more often observed along the periphery of the tumor nodes of these mice, and multiple clusters of atypical cells dying by cytolysis are detected in the central areas of the tumor parenchyma. (Table 1).

In depigmented melanoma sections from T1023-treated mice in the peritumoral area, the content of CD31-positively stained endothelial cell nuclei and the quantitative density of vessels were statistically significantly lower than in tumors from the control and Na-DCA-treated mice (Figure 6; Table 2). Along the periphery of tumor nodes in T1023-treated mice, an uneven decrease in the intensity of immunostaining of the nuclei of proliferating cells for Ki-67 was determined, and the proportion of Ki-67-positive nuclei of atypical cells in these tumors was significantly lower (74%) than in the control (Figure 7; Table 3), and, on the contrary, the number of immunopositive tumor cells for Caspase-3 significantly increased, more than two times (23–29 per mm^2^) in comparison with the control (Figure 8; Table 4).

Upon macroscopic examination, the volume of tumor nodes of melanoma B-16 in T1084-treated mice was noticeably 1.2–1.6 times lower than in mice of other groups, and necrotic and dystrophic changes in them were most pronounced (Figure 5A). In the peripheral growth zones of these tumors, a decrease in the content of blood vessels was also clearly visualized, as was the case with exposure to T1023. In areas of solid parenchyma structure, the vessels are distributed unevenly. H&E staining often revealed eosinophilic contours of destroyed vessels. The central zone of tumor nodes in T1084-treated mice is represented by extensive oxyphilic fields of spontaneous necrosis, reaching the periphery of the tumor. In some tumor nodes, the expansion of central necrosis is noted, which in some places involved a significant part of the neoplastic tissue. In the peripheral parts of the tumors, the foci of necrosis with local hemorrhages also look more extensive and branched than in other groups. Against the background of T1084 injections, the polymorphism of tumor cells increases in the melanoma parenchyma (Figure 5B,C). Along with the expanded perinecrotic rims of atypical, immunonegative small cells in a state of dystrophy, extensive diffuse foci with an uneven intensity of the staining of tumor cells for the studied markers are clearly visible in areas of the solid structure of the parenchyma. According to quantitative analysis, the content of viable parenchyma in tumor nodes of T1084-treatel mice was 69%, which was significantly lower than in the control and in mice exposed to Na-DCA or T1023 (Table 1).

As a result of an immunohistochemical reaction for CD31 in depigmented melanoma sections from T1084-treated mice, only individual fragments of blood vessels were stained. In the peritumoral area, the content of CD31-positively stained endothelial cell nuclei and the quantitative density of blood vessels were statistically significantly lower than in tumors from the control and Na-DCA-treated mice (Figure 6; Table 2). In areas of the parenchyma occupied by small tumor cells with dystrophic changes, the weak new formation of microvasculature vessels was noted. Along the periphery of tumor nodes in T1084-treated mice, a pronounced decrease in the intensity of immunostaining of the nuclei of proliferating cells for Ki-67 was determined, and the proportion of Ki-67-positive nuclei of tumor cells in these tumors was significantly lower (67%) than in the control (Figure 7; Table 3). Tumor cells with nuclei intensely stained for Caspase-3 were more often present at the periphery of the tumor near the vessels of the peritumoral region, around cellular debris, in tumor cuffs, and with invasive growth into adjacent muscle tissue (Figure 8). The number of immunopositive tumor cells for Caspase-3 in tumor nodes in T1084-treated mice increased significantly, more than three times (27–36 per mm^2^) compared to the control (Table 4).

## 3. Discussion

In the last thirty years, much attention has been paid to the study of the nature of tumor-acquired resistance to anti-angiogenic therapy. Currently, this is an extensive area of knowledge that describes a significant number of mechanisms for reducing tumor reactivity and adaptation to anti-angiogenic stress, caused by the multifaceted interaction of genetic, epigenetic, and microenvironmental factors [3,4,6,7,8]. At the same time, many authors note the important role of the hypoxia-dependent pathways of neoplasia adaptation and metabolic reprogramming of tumor cells in these processes [41,42,43,44,45,46,47].

In this regard, to prevent the development of resistance of solid tumors to the anti-angiogenic action of the NOS inhibitor T1023 (1-isobutanoyl-2-isopropylisothiourea hydrobromide; Figure 1A), we considered it appropriate to combine this agent with the PDK inhibitor dichloroacetate (DCA; Figure 1B), the toxic effect of which is focused on the mitochondria of hypoxic cells [30,31,48,49]. Thus, the new compound T1084 contains in its molecular structure both NOS-inhibiting and PDK-inhibiting fragments: 1-isobutanoyl-2-isopropylisothiourea dichloroacetate salt (Figure 1C) [39]; and this compound is a bifunctional NOS/PDK inhibitor [40].

The aim of this study was to investigate the features of the antitumor activity of the NOS/PDK inhibitor T1084. Initially, a comparative assessment of the cytotoxic activity in vitro of T1023 and T1084 using the MTT-test with six human malignant cell lines (A375, MFC-7, K562, HeLa, OAW42, and PC-3) and one normal human cell line (MRC-5) (Figure 2) was conducted. The data obtained indicated a similar, statistically equal cytotoxic effect of T1023 and T1084 on the majority of cell lines (A375, MFC-7, K562, OAW42, PC-3, and MRC-5). The cytotoxicity of T1023 and 1084 against most of these cell lines (MFC-7, K562, OAW42, PC-3, and MRC-5) is of a nonspecific toxic nature. The IC_50_ values obtained on these cells (0.65–1.41 mM) correspond to subtoxic and toxic doses (more than 0.7×LD_16_) of these compounds in the in vivo acute lethality test [40].

The human melanoma A375 cell line became the exception in this series. A375 cells showed 2–2.5 times higher sensitivity to the cytotoxic effects of both T1023 and T1084 than other cell lines studied. In this case, the IC_50_ values (0.39–0.41 mM) were at a safer toxicological level (about 0.4×LD_16_), and the high values of the selectivity index (the ratio between the IC_50_ values obtained on MRC-5 and malignant cells) of the A375 cells (2.2–2.4) suggested the presence of the specific cytotoxicity of both T1023 and T1084 to melanoma. The nature of the relatively high sensitivity of melanoma to the NOS-inhibitory action of T1023 and T1084 in vivo may be associated with the high role of NO and NOS expression not only in angiogenesis, but also in the regulation of proliferative activity and resistance to apoptosis of cells of this neoplasia [50,51,52,53].

This study also brought attention to the human cervical cancer HeLa cell line. HeLa cells showed low sensitivity to the cytotoxicity of T1023 (IC_50_: 0.81 mM), but the cytotoxic effect of T1084 against these cells was significantly higher (IC_50_: 0.54 mM). The high value of the T1084 selectivity index for HeLa cells (1.7) also suggests the presence of the specific cytotoxicity of NOS/PDK inhibitor T1084 against cervical cancer.

These results justified the antitumor effects of T1084 and the precursor compounds, Na-DCA and T1023, in two solid tumor models in vivo: cervical cancer CC-5 and melanoma B-16 in mice. All in vivo experiments were performed according to the general scheme. Experimental treatment began on the seventh–eighth day after tumor inoculation. From this day until the 20th day of tumor growth, the mice of the experimental groups were injected i.p. daily with T1023, Na-DCA, or T1084 at equimolar doses of 0.22 mmol/kg (60.0, 33.6, and 70.7 mg/kg, respectively). The mice of the control groups were injected i.p. daily with 0.9% sodium chloride in an equivalent volume.

The results of the study on the cervical cancer CC-5 model showed (Figure 3) that this neoplasia was initially highly sensitive to the Na-DCA or T1023 action. After just two injections of these agents, reliable antitumor effects were observed both in the Na-DCA-treated and T1023-treated mice groups, accompanied by significant (40%) inhibition of tumor growth. But then CC-5 began to quickly and effectively evade the subsequent effects of the NOS- and PDK inhibitors. Already from the fifth day of treatment and until the end of observation, the antitumor effects of both Na-DCA and T1023 in mice of these groups had reached an insignificant level. In contrast, in two independent experiments, no significant morphological manifestations of CC-5 adaptation to the action of the NOS/PDK inhibitor T1084 were observed. In both experiments, CC-5 retained high sensitivity to T1084 subchronic exposure. A pronounced reliable antitumor effect, accompanied by the stable inhibition (by 43–58%) of tumor growth, was registered throughout the entire observation period.

The results of the study on melanoma B-16 showed (Figure 4) the poor sensitivity of this neoplasia to subchronic exposure to Na-DCA at the dose used. No significant antitumor effects were observed in Na-DCA-treated mice throughout the experiment, and tumor growth inhibition in this group was weak (less than 20%). The effect of Na-DCA on the morphology of melanoma tumor nodes after five injections of this agent was also quite weak. A slight increase in necrotic and dystrophic changes in tumors of Na-DCA-treated mice (Figure 5; Table 1), a moderate increase in the uneven distribution of vessels (Figure 6; Table 2), and a tendency towards increased apoptosis of tumor cells (Figure 8; Table 4) were noted. At the same time, a significant decrease (1.6 times) in the proliferative activity of tumor cells was observed in these tumors (Figure 7; Table 3).

Melanoma B-16 was more sensitive to subchronic exposure to NOS inhibitor T1023 (Figure 4). Its antitumor effect gradually increased during the course of administration and reached a maximum (tumor growth inhibition—30–33%) after seven–nine injections. However, in the later stages of the experiment, a tendency towards the development of neoplasia resistance was noted—the effect was noticeably weakened (tumor growth inhibition—27%). The effect of T1023 on the morphology of melanoma tumor nodes was also pronounced after five injections. Histological data indicated that, as in studies on Lewis lung carcinoma [24,25], its influence on melanoma was anti-angiogenic in nature. In the peritumoral region of the tumor nodes of T1023-treated mice, a significant decrease in the content of endothelial structures and the quantitative density of vessels was observed; as well as in the tumor parenchyma, a pronounced uneven distribution of microvessels was noted and the contours of destroyed vessels were visualized (Figure 6; Table 2). The consequence of significant vascular disturbances was a significant decrease in the content of viable parenchyma in T1023-treated mice tumors due to an almost 2-fold expansion of the zones of dystrophic, immunonegative cells (Figure 5; Table 1), as well as a significant decrease (by 43%) in proliferative activity (Figure 7; Table 3) and a significant (2.4-fold) increase in apoptotic death of tumor cells (Figure 8; Table 4).

At the same time, melanoma B-16 was consistently highly sensitive to subchronic exposure to the NOS/PDK inhibitor T1084 (Figure 4). The antitumor effect of T1084 developed rapidly and increased throughout the course of agent administration. A significant effect of T1084 was observed just after two injections and reached its maximum after nine–thirteen injections (tumor growth inhibition—59–62%). In this case, no morphological manifestations of neoplasia adaptation to T1084 treatment were observed. Histological data indicated that T1084 realized an anti-angiogenic effect on melanoma similar to that of T1023 (Figure 6; Table 2). However, the severity of pathomorphological changes in melanoma tumor nodes in T1084-treated mice was qualitatively higher. In these tumors, the content of viable parenchyma was significantly lower than in all other groups, due to a 3-fold expansion of necrotic and dystrophic zones (Figure 5; Table 1). The action of T1084 was also accompanied by a significant decrease (by 53%) in proliferative activity (Figure 7; Table 3) and a significant (2.5-fold) increase in the apoptotic death of tumor cells (Figure 8; Table 4).

Thus, the combined data obtained in this in vivo study on cervical cancer CC-5 and melanoma B-16, as well as the data obtained in our first study of T1084 on solid Ehrlich carcinoma [40], indicate that the pharmacodynamics of T1084 antitumor action develop through the interaction of NOS-dependent and PDK-dependent pathophysiological effects of this NOS/PDK inhibitor. The NOS inhibitory activity of T1084, like that of T1023, mediates anti-angiogenic effects on neoplasia. At the same time, the PDK inhibitory activity of T1084 enhances the cytotoxicity of induced intratumoral hypoxia, and suppresses the development of neoplasia adaptation to anti-angiogenic stress. Such pharmacodynamic features allow compound T1084, when administered subchronically in safe doses (1/5 LD_16_), to overcome tumor resistance and achieve a stable synergistic antitumor effect.

In conclusion, it should be noted that the chemical-pharmacological approach that we used in developing the T1084 compound potentially allows the development of other multifunctional antitumor agents with different mechanisms of action. In particular, we succeeded in synthesizing NOS-inhibiting isothiourea salts with 3-bromopyruvate (HK2 inhibitor) and α-cyano-4-hydroxycinnamate (MCT inhibitor), which also exhibit synergistic antitumor activity [54,55,56], and which we are currently studying.

Overall, this rather limited study leaves open a lot of questions that determine the potential of the T1084 compound. The nature of the differences in neoplastic susceptibility to this NOS/PDK inhibitor is not fully understood. The toxicological acceptability of this agent for long-term treatment has not been studied. The potential of T1084 in combination therapy with radiation and with alkylating and antiproliferative chemotherapy drugs, as well as with antitumor immunomodulators, has not been studied. We intend to pay due attention to these issues in the near future.

## 4. Materials and Methods

### 4.1. T1084 and Other Compounds

The studied compound T1084—1-isobutanoyl-2-isopropylisothiourea dichloroacetate, and the substances used for a comparison—1-isobutanoyl-2-isopropylisothiourea hydrobromide (T1023) and sodium dichloroacetate (Na-DCA), were synthesized in the laboratory of radiation pharmacology A. Tsyb MRRC. T1023 was synthesized by the previously described method [57]. The method for T1084 preparation consisted of the isolation of the free base, 1-isobutanoyl-2-isopropylisothiourea, from the T1023 compound, which was then reacted with DCA [39,40]. The synthesis of Na-DCA was carried out by the interaction of equimolar amounts of DCA and NaOH.

The structures of T1023 and T1084 were confirmed by NMR spectrometry and elemental analysis. The spectra ^1^H NMR and ^13^C NMR were obtained in DMSO-d6 on the Fourier spectrometer AVANCE AV 300 (Bruker, Ettlingen, Germany). The elemental analysis of C, H, and N was performed on an elemental analyzer EA 1108 (Carlo Erba EA 1108 elemental analyser, Carlo Erba Instruments, Milano, Italy). The control of substances’ purity was performed using thin layer chromatography, high-performance liquid chromatography, and melting temperature measurements. Thin layer chromatography on Silufol UV-254 plates (Kavalier, Votice, Czech Republic) was performed in a benzene–ethanol–triethylamine (9:1:0.1) system. Quantitative determination of the content of the active substance and impurities was carried out using high-performance liquid chromatography on a Hitachi Chromaster HPLC System (Hitachi High-Tech Corp., Santa Clara, CA, USA). The melting point was determined on an automatic heating unit PTP-M (JSC “Laboratory Equipment and Instruments” (LOIP, Ltd.), Saint Petersburg, Russia). The laboratory methods used for the synthesis, isolation, and purification of T1084, T1023, and Na-DCA provided stable quality substances with an active ingredient content of 97–98% and a total impurities content of less than 1% of dry weight.

### 4.2. Cytotoxic Activity

#### 4.2.1. Cell Lines

A normal human fetal lung fibroblast cell line (MRC-5), human melanoma cell line (A375), human cervix adenocarcinoma cell line (HeLa), human mammary ductal adenocarcinoma cell line (MFC-7), human chronic myelogenous leukemia cell line (K562), human prostatic adenocarcinoma cell line (PC-3) (American Type Culture Collection, Manassas, VA, USA), and a human ovarian carcinoma cell line (OAW42) (Federal Scientific and Clinical Center of Federal Biomedical Agency of Russia, Moscow) were grown in RPMI-1640 medium (PanEco, Ltd., Moscow, Russia). The medium was supplemented with 10% fetal bovine serum, L-glutamine, and penicillin-streptomycin (PanEco Ltd., Moscow, Russia). Cells were grown at 37 °C in a humidified atmosphere with 5% CO_2_ in the air.

#### 4.2.2. Treatment of Cell Lines

Stock solutions (100 mg/mL) of compounds T1023 and T1084 diluted in dimethyl sulfoxide (DMSO; PanEco Ltd.) were dissolved in the corresponding media to the required working concentrations. Target HeLa (2000 cells per well), A375 (5000 cells per well), MCF-5 (5000 cells per well), K562 (5000 cells per well), PC-3 (5000 cells per well), OAW42 (5000 cells per well), and MRC-5 (7000 cells per well) cells were seeded into the wells of 96-well flat-bottomed microtiter plates. After cell adherence for 24 h, five double-diluted concentrations of the investigated compounds were added to the wells. The final concentrations applied to target cells were 1000, 500, 250, 125, and 62.5 μg/mL, except in the control wells, where only the nutrient medium was added to the cells. The final concentrations of the DMSO solvent did not exceed 0.5% and were nontoxic to cells. The cell cultures were incubated for 72 h.

#### 4.2.3. Determination of Cell Survival

The effects of the compounds T1023 and T1084 on cell survival were determined by the MTT test [58]. Briefly, the (3-(4,5-di-methyl-thiazol-2-yl)-2,5-diphenyl tetrazolium bromide (MTT; PanEco Ltd.) was dissolved (5 mg/mL) in deionized water (PanEco) and filtered (0.22 μm) before use. An amount of 20 μL of MTT solution was added to each wall. Samples were incubated for 4 h at 37 °C in 5% CO_2_ and a humidified air atmosphere. Then, the culture medium in the wells was replaced with 100 μL of DMSO to extract insoluble formazan resulting from the conversion of the MTT dye by viable cells. The number of viable cells in each well was proportional to the intensity of the absorbance (A) of light, which was then read in an StatFax 2100 plate reader (Awareness Technology, Inc., Palm City, FL, USA) at 492 nm (reference wavelength—630 nm). To determine cell survival (%), the A of a sample with cells grown in the presence of various concentrations of compounds T1023 or T1084 was divided by the control optical density (the A of the control cells grown only in nutrient medium) and multiplied by 100. In each experiment, the A of the blank was always subtracted from the A of the corresponding sample with target cells. IC_50_ was defined as the concentration of an agent inhibiting cell survival by 50% compared with the vehicle-treated control. IC_50_ values for T1023 and T1084 were calculated in units of mM, taking into account the molecular weight of these compounds (269.2 and 317.2 g/mol, respectively [40,57]). All experiments were performed in quadruplicate. The data were collected from four independent experiments and presented as mean ± SDs.

The obtained concentration–survival data were analyzed using the logistic model of the Origin 6 software package (“Non-linear curve fit—Logistic”) and the concentration (in µg/mL units) at which the half-maximal inhibition of cell growth was observed (IC50) was calculated. The obtained IC_50_ values for T1023 and T1084 were then converted to mM units by dividing by the corresponding molecular weight (this is described above in Section 4.2.3). A table (Appendix A) reflecting the IC_50_ estimates for all cell lines obtained in independent experiments is provided in Appendix A.

### 4.3. In Vivo Antitumor Activity

#### 4.3.1. Animals

Female mice CBA (2 months old, 19–22 g) and female mice C57Bl/6j (2–2.5 months old, 18–23 g) were used in these studies. The animals were purchased from the Biomedical Technology Scientific Center of Federal Biomedical Agency of Russia (Moscow). Mice were kept in the vivarium of the A. Tsyb MRRC in T-3 and T-4 cages under natural light conditions with forced ventilation 16 times/h, at a room temperature of 18–20 °C and relative humidity of 40–70%. The animals had free access to water and PK-120-1 rodent feed (Laboratorkorm, Ltd., Moscow, Russia). The work with laboratory animals was approved by the MRRC Ethical Commission (protocol number: 1-D-00041) and was performed in accordance with generally accepted norms of animal handling based on standard operating procedures adopted at the A. Tsyb MRRC and corresponding to the rules and requirements of European Convention ETS/STE No. 123 and the international GLP standard.

#### 4.3.2. Tumor Models

Two transplantable mouse tumors were used: cervical cancer (CC-5) and melanoma (B-16). Strain mice cervical cancer CC-5 was obtained from the tumor bank of the N.N. Blokhin National Medical Research Center for Oncology of the Ministry of Health of Russia, and was maintained on female CBA mice. The mice melanoma B-16 cells (Culture collection of P.A. Herzen Moscow Oncology Institute) were grown in DMEM medium (PanEco Ltd.) supplemented with 10% fetal bovine serum, L-glutamine, and penicillin-streptomycin (PanEco Ltd.), at 37 °C in a humidified atmosphere with 5% CO_2_ in the air. Before tumor transplantation, the hair on the lateral surface of the right thigh of the mice was removed using Trimmer ChroMini Type 1591 (Moser, Germany). The CC-5 strain was transplanted into female CBA mice by the subcutaneous injection of 2.5 × 10^6^ tumor cells in 100 μL of the medium 199 (PanEco Ltd.) to the lateral surface of the right thigh. The B-16 strain was transplanted into female C57Bl/6j mice by the subcutaneous injection of 10^6^ tumor cells in 100 μL of the medium DMEM to the lateral surface of the right thigh.

#### 4.3.3. Compounds and Solutions Used in In Vivo Experiments

The studied compound T1084 and the substances used for a comparison, T1023 and Na-DCA, were used in the form of 0.71%, 0.60%, and 0.34% solutions, respectively. All injectable solutions were prepared *ex tempore* with water for injection (Dalchimpharm, JSC, Khabarovsk, Russia). In all experiments, solutions of T1084, T1023, and Na-DCA were administered to the animals daily intraperitoneally (i.p.) in a volume of 10 mL/kg, which provided equimolar doses of 0.22 mmol/kg of these compounds—70.7, 60.0, and 33.6 mg/kg, respectively. The animals in the control groups received daily i.p. injection of an equivalent volume of a 0.9% sodium chloride solution (Dalchimpharm JSC, Russia).

#### 4.3.4. Treatment of Mice Tumors

Three independent experiments were performed on these tumor models using similar schemes: two on mice with cervical cancer CC-5 and one on mice with melanoma B-16. Briefly, after the transplantation of neoplasia, tumor-bearing mice were randomized into groups of equal size. Experimental treatment began on the 7th–8th day after inoculation, when the tumor nodes in all individuals reached a reliably measurable size: 60–80 mm^3^. From this day until the 20th day of tumor growth, mice of the experimental groups were i.p. injected daily with T1023, Na-DCA, or T1084 at equimolar doses of 0.22 mmol/kg (60.0, 33.6, and 70.7 mg/kg, respectively) in solution volume 10.0 mL/kg. Mice of the control groups were injected daily with 0.9% sodium chloride solution in volume 10.0 mL/kg. The duration of the experiments was 21–24 days from the tumor transplantation.

The first experiment on the model of cervical cancer CC-5 was carried out on 80 female CBA mice, divided into a control and three experimental groups (20 mice in each). Daily exposure of mice to all studied agents was carried out from the 7th to 20th days of tumor growth. The total duration of observation was 23 days after tumor transplantation. The second experiment on the model of cervical cancer CC-5 was carried out on 40 female CBA mice, divided into control and experimental groups (20 mice in each). Daily i.p. injections of 0.9% sodium chloride and T1084 were carried out from the 8th to 20th days of tumor growth. The total duration of observation was 24 days after tumor transplantation. The experiment on B-16 melanoma was carried out on 88 female C57Bl/6j mice, divided into a control and three experimental groups (22 mice in each). Daily exposure of mice to all studied agents was carried out from the 8th to 20th days of tumor growth. The total duration of observation was 21 days after tumor transplantation. In this experiment, on the 13th day of melanoma growth (6th day from the beginning of the exposure), 7 mice from each group were bred for pathomorphological and histological studies. 

#### 4.3.5. Morphological Assessment of Effects

The development of neoplasia in experimental groups of mice was assessed morphometrically. For this purpose, the size of tumor nodes in all mice was measured every 2–3 days using a digital caliper SCC-1-125 (RPE Chiz, Ltd., Chelyabinsk, Russia), and their volumes (*TV*) were estimated in elliptical approximation: TV=a⋅b⋅c×(π/6), where a, b, and c are orthogonal diameters. Then, the relative volumes of the neoplasias were calculated by normalizing the tumor volume in each animal on the day of the beginning of the exposure. In addition, the indexes of inhibition of tumor growth were calculated for all treated mice: $TI_{i,t}=\left({\bar{TV}}_{C,t}-TV_{i,t}\right)/{\bar{TV}}_{C,t}\times 100\%$, where $TI_{i,t}$ is the inhibition index in the *i*th animal at a time of observation *t*; ${\bar{TV}}_{C,t}$ is the average relative tumor volume in control at the time of observation *t*; $TV_{i,t}$ is relative tumor volume of the *i*th animal at a time of observation *t*. The severity and dynamics of the Na-DCA, T1023, and T1084 antitumor effect were assessed and compared by means of multiple intergroup statistical comparisons of the relative tumor volumes and indicators of tumor growth inhibition at different periods of observation.

Integral antitumor effects were assessed by the duration of tumor growth inhibition and the weight of tumor nodes in groups at the final stage of observation. The duration of tumor growth delay (in days) was assessed graphically using growth curves over the time of a 10-fold (for CC-5 cervical cancer) and 6-fold (for B-16 melanoma) increase in tumor volume. At the end of the experiment, tumor nodes were isolated under thiopental anesthesia (Sintez, OJS, Kurgan, Russia). Tumor weights were measured on an AR-0640 analytical balance (Ohaus Europe, Nänikon, Switzerland).

#### 4.3.6. Histological and Immunohistochemical Assessment of Effects

A comparative study of the effect of Na-DCA, T1023, and T1084 on tumor morphology was carried out on B-16 melanoma. For this, at the exponential stage of neoplasia growth (on the 13th day after transplantation; on the 6th day after the beginning of exposure), 7 tumor-bearing mice were taken from each group. After euthanasia under thiopental anesthesia, tumor nodes were isolated. Tumor tissue in the form of plates 3–4 mm thick, oriented along the long axis, was fixed in a buffered formalin solution (Biovitrum, Ltd., St. Petersburg, Russia). After standard histological processing on a Leica TP1020 carousel histoprocessor (Leica Biosystems Inc., Buffalo Grove, IL, USA), tissue samples were dehydrated and embedded in the Histomix paraffin medium (Sakura Finetek USA, Torrance, CA, USA) at the HistoStar filling station (Thermo Fisher Scientific, Waltham, MA, USA). For morphological studies, sections with a thickness of 5 μm, obtained with a Leica RM2235 microtome (Leica Biosystems Inc.), were stained with hematoxylin and eosin (Biovitrum Ltd., Cambridge, UK) after deparaffinization. The general histology of B16 melanoma was studied in accordance with standard techniques using a Leica DM 1000 microscope (Leica Microsystems CMS GmbH, Wetzlar, Germany) with microphotography using a Leica ICC50 HD digital camera (Leica Microsystems, CMS GmbH, Wetzlar, Germany).

Immunohistochemical studies on serial sections 5 µm thick were carried out using the biotin–streptavidin–peroxidase complex method using polyclonal rabbit antibodies to Ki67 (1:200, ThermoFisher, USA) and monoclonal rabbit antibodies to Caspase-3 (EPR18297, 1:1000, Abcam, Waltham, MA, USA) and to CD31, (EPR17259, 1:250, Abcam, USA) according to the standard protocol. Before applying primary antibodies, sections were depigmented in 3% hydrogen peroxide in PBS (PanEco Ltd.) in a thermostat for 2 h at 55 °C. In a solution of antibodies to Ki67, CD31, and Caspase-3, the preparations were incubated overnight in a humid chamber at 4 °C. To reduce the level of nonspecific reactions, a protein block (X0909, Dako, Denmark) was used. After washing in PBS (pH 7.4), the material was treated with secondary antibodies using biotinylated goat anti-rabbit IgG antibodies (ab205718, 1:500, Abcam, USA). Substrate peroxidase was expressed by diaminobenzidine (Liquid DAB+, Dako).

Integrated morphometric indicators, including the volumetric content of the parenchyma of neoplasms and zones of necrosis, the quantitative density of cells in the parenchyma, the proportion of tumor cells with a reaction of cell nuclei to Ki-67 and Caspase-3, and the quantitative characteristics of the angioarchitecture of melanoma were analyzed on digital scans of histological preparations (Leica Microsystems ScanScope CS2, Germany). Quantitative analysis was carried out using QuPath v.0.5.1 software [59] and ImageJ 1.54f (Wayne Rasband and contributors, National Institutes of Health, Stapleton, NY, USA) according to the basic principles of stereology in morphometry. The quantitative densities of cell nuclei based on the number of their sections (cellularity), apoptosis index, and mitotic index in the growth zones of tumor nodes were determined according to the standard method, using two sections of the studied tumor. Proliferating cells were identified by positive staining for Ki-67. Cells dying by apoptosis were identified by immunohistochemical staining for Caspase-3. The total test area for each tumor was at least 1.25 mm^2^.

The angioarchitecture of melanoma was assessed according to the principles described in [60,61,62]. Quantitative indicators, including the volumetric content of endothelial structures and the density of microvessels and their cross-sectional area, were determined in the peritumoral area and in the “hot spots” of the parenchyma, with the most intense neovascularization. The peritumoral area, 0.5 mm wide, included the peripheral edge of the tumor and subcutaneous tissue. The total peritumoral area tested for each tumor was at least 1.25 mm^2^. To determine the characteristics of vessels in the melanoma parenchyma, 20 “hot spots” were analyzed in the fields of view of each tumor section.

### 4.4. Statistical Analysis

The standard parameters of the variation statistics were calculated for all experimental data. Their values are given, including graphically, in the form M ± SD. The level of significance of intergroup differences in the indicators was assessed using nonparametric statistical means. For pairwise comparisons, the Mann–Whitney U test was used. For multiple comparisons, we used the Kruskal–Wallis ANOVA of rank with the post hoc Mann–Whitney U test with Holm–Bonferroni corrections [63]. In all cases, the effects and differences were considered statistically significant at the 5% level. Statistical calculations were performed using the software package Statistica 10.0 (StatSoft Inc., Tulsa, OK, USA) and BioStat 7.3 (AnalystSoft Inc., Alexandria, CA, USA).

## Figures and Tables

**Figure 1 ijms-25-09711-f001:**
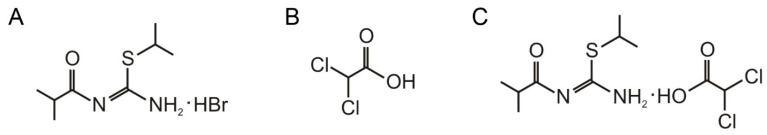
Structural formulas of the compounds studied. (**A**) 1-isobutanoyl-2-isopropylisothiourea hydrobromide (compound T1023); (**B**) dichloroacetate (DCA); (**C**) 1-isobutanoyl-2-isopropylisothiourea dichloroacetate (compound T1084).

**Figure 2 ijms-25-09711-f002:**
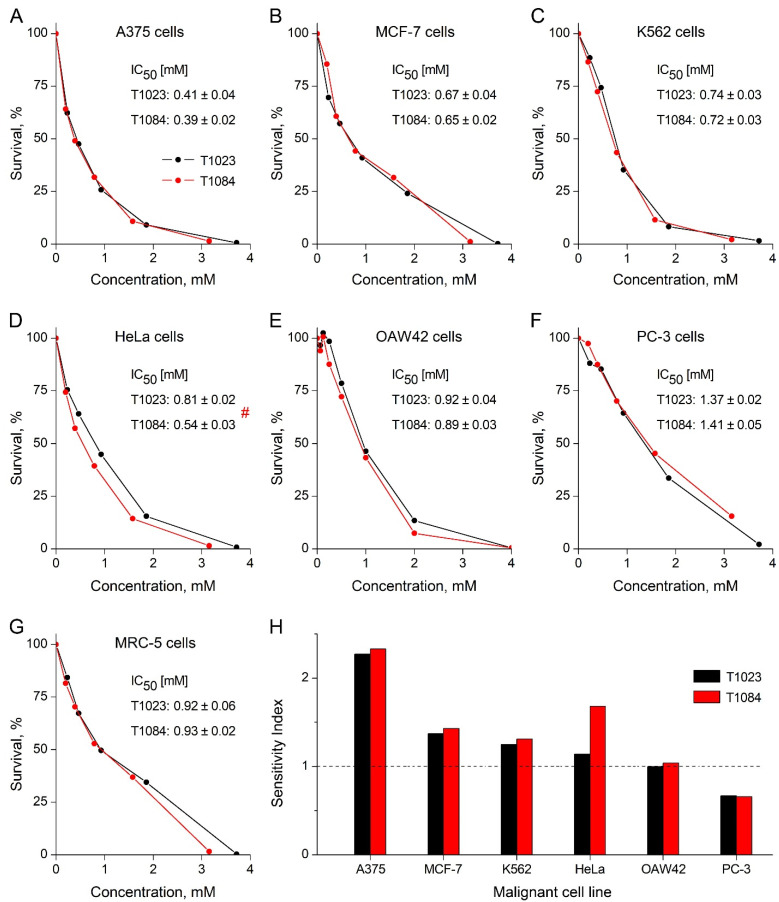
Cytotoxic effect of compounds T1023 and T1084 in vitro according to the MTT-test in relation to human malignant and normal cells. (**A**–**G**) Representative graphs of survival of human melanoma A375 cells (**A**), human mammary ductal adenocarcinoma MFC-7 cells (**B**), human chronic myelogenous leukemia K562 cells (**C**), human cervical cancer HeLa cells (**D**), human ovarian carcinoma OAW42 cells (**E**), human prostate cancer PC-3 cells (**F**), and a normal human fetal lung fibroblast MRC-5 cells (**G**) after 72 h of cell growth in the presence of increasing concentrations of T1023 and T1084. IC_50_ values were determined as M ± SD from four independent experiments. # significant difference (*p* = 0.01764) between the IC_50_ values for T1023 and T1084. (**H**) Sensitivity indexes of malignant cell lines to toxic effects of T1023 and T1084, calculated as the ratio of the IC_50_ value for MRC-5 and the IC_50_ value for the corresponding cell line.

**Figure 3 ijms-25-09711-f003:**
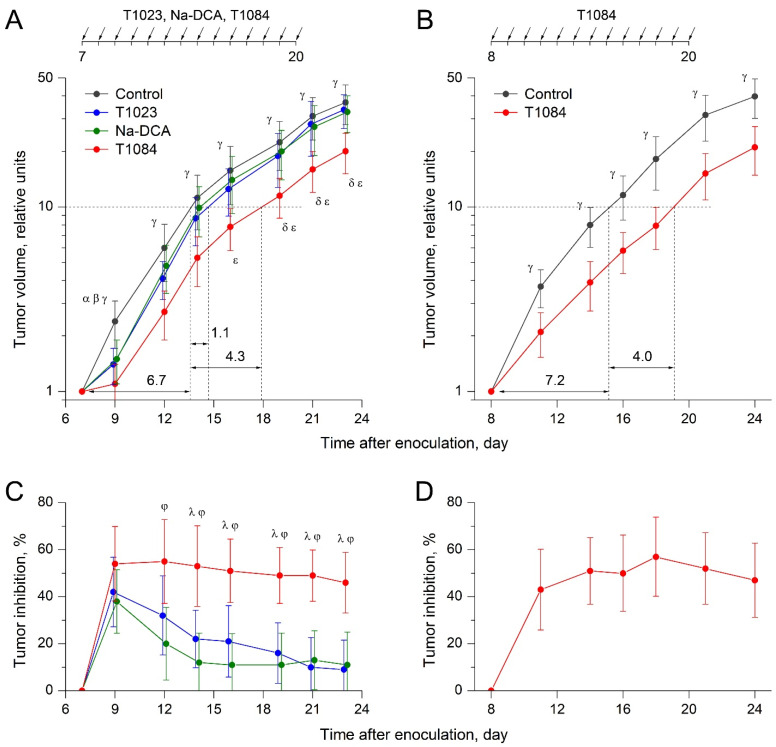
Effect of T1023, Na-DCA, and T1084 with chronic i.p. administration at equimolar doses 220 μM (60.0, 33.6, and 70.7 mg/kg, respectively) on cervical cancer CC-5 in female mice CBA. Data from two independent experiments ((**A**,**C**)—first; (**B**,**D**)—second). (**A**,**B**) Tumor growth curves in animal groups. Tumor volume (TV) indexes for each animal are normalized to the initial volume on the day of the beginning of treatment. Graphical deviations correspond to SD (*n* = 16–20 per point). Dotted lines—estimates of the duration of tumor growth delay (in days) by the time of a 10-fold increment in TV. Symbols—significant TV differences: α—in control mice vs. T1023-treated mice; β—in control mice vs. Na-DCA-treated mice; γ—in control mice vs. T1084-treated mice; δ—in T1084-treated mice vs. T1023-treated mice; ε—in T1084-treated mice vs. Na-DCA-treated mice. Significance of differences: α—(**A**) *p* = 0.00953; β—(**A**) *p* = 0.01124; γ—(**A**) *p* = 0.00345, *p* = 0.00052, *p* = 0.00496, *p* = 0.00521, *p* = 0.00370, *p* = 0.00128, *p* = 0.00039; (**B**) *p* = 0.00630, *p* = 0.00415, *p* = 0.00172, *p* = 0.00074, *p* = 0.00258, *p* = 0.00523; δ—(**A**) *p* = 0.01045, *p* = 0.00438, *p* = 0.00272; ε—(**A**) *p* = 0.00877, *p* = 0.00739, *p* = 0.00843, *p* = 0.00946. (**C**,**D**) Dynamics of tumor growth inhibition (TI) in animal groups. Graphical deviations correspond to SD. Symbols—significant TI differences: λ—in T1084-treated mice vs. T1023-treated mice; φ—in T1084-treated mice vs. Na-DCA-treated mice. Significance of differences: λ—(**C**) *p* = 0.01443, *p* = 0.01319, *p* = 0.00557, *p* = 0.00072, *p* = 0.00094; φ—(**C**) *p* = 0.01028, *p* = 0.00813, *p* = 0.00776, *p* = 0.00319, *p* = 0.00174, *p* = 0.00115.

**Figure 4 ijms-25-09711-f004:**
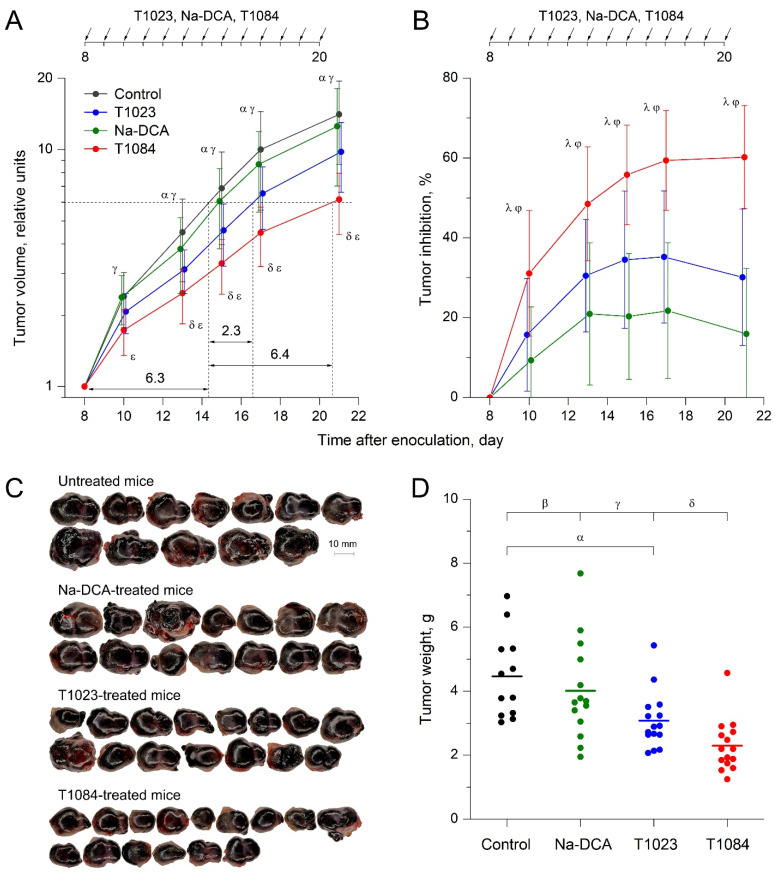
Effect of T1023, Na-DCA, and T1084 with chronic i.p. administration at equimolar doses 220 μM (60.0, 33.6, and 70.7 mg/kg, respectively) on melanoma B-16 in female mice C57Bl/6j. (**A**) Tumor growth curves in animal groups. Tumor volume (TV) indexes for each animal are normalized to the initial volume on the day of the beginning of treatment. Graphical deviations correspond to SD (*n* = 12–20 per point). Dotted lines—estimates of the duration of tumor growth delay (in days) by the time of a 6-fold increment in TV. Symbols—significant TV differences: α—in control mice vs. T1023-treated mice; γ—in control mice vs. T1084-treated mice; δ—in T1084-treated mice vs. T1023-treated mice; ε—in T1084-treated mice vs. Na-DCA-treated mice. Significance of differences: α—*p* = 0.03196, *p* = 0.00950, *p* = 0.00679, *p* = 0.03592; γ—*p* = 0.00395, *p* = 0.00062, *p* = 0.00002, *p* = 0.00007, *p* = 0.00014; δ—*p* = 0.02783, *p* = 0.01080, *p* = 0.00289, *p* = 0.00230; ε—*p* = 0.00118, *p* = 0.00234, *p* = 0.00004, *p* = 0.00013, *p* = 0.00036. (**B**) Dynamics of tumor growth inhibition (TI) in animal groups. Graphical deviations correspond to SD. Symbols—significant TI differences: λ—in T1084-treated mice vs. T1023-treated mice; φ—in T1084-treated mice vs. Na-DCA-treated mice. Significance of differences: λ—*p* = 0.03189, *p* = 0.01854, *p* = 0.00540, *p* = 0.00144, *p* = 0.00115; φ—*p* = 0.00056, *p* = 0.00139, *p* = 0.00002, *p* = 0.00008, *p* = 0.00021. (**C**) Appearance of melanoma B-16 tumor nodes on the 21th day after inoculation in animal groups. (**D**) Weight of melanoma B-16 tumor nodes on the 21th day after inoculation in animal groups (*n* = 12–15). Symbols—significant tumor weight differences: α—in T1023-treated mice vs. control mice (*p* = 0.01163); β—in T1084-treated mice vs. control mice (*p* = 0.00006); γ—in T1084-treated mice vs. Na-DCA-treated mice (*p* = 0.00306); δ—in T1084-treated mice vs. T1023-treated mice (*p* = 0.03032).

**Figure 5 ijms-25-09711-f005:**
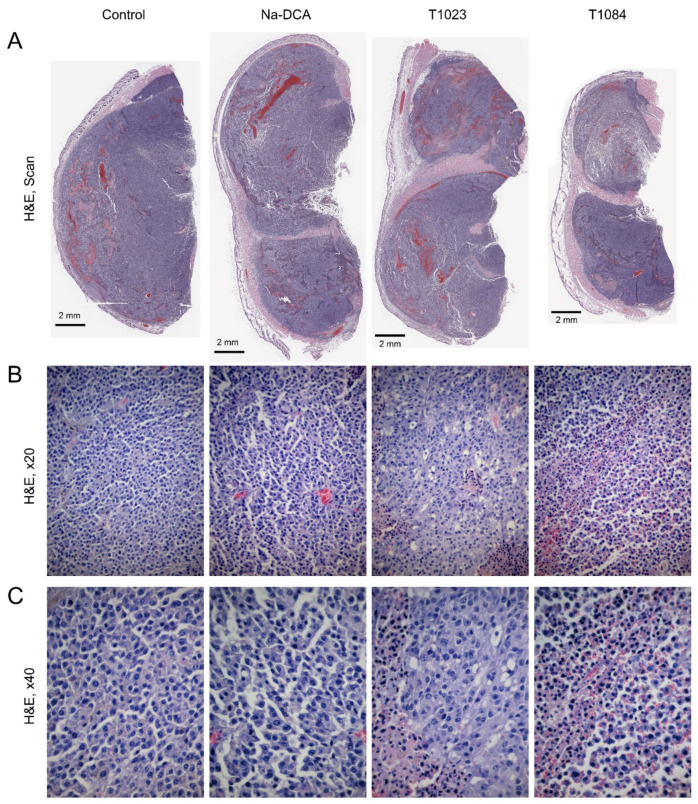
Effect of T1023, Na-DCA, and T1084 with chronic i.p. administration at equimolar doses 220 μM (60.0, 33.6, and 70.7 mg/kg, respectively) on the general histology of B-16 melanoma in female mice C57Bl/6j on the 6th day after the beginning of exposure. Hematoxylin and eosin staining. (**A**) Review preparations of tumor nodes. (**B**,**C**) Parenchyma of tumor nodes, objective magnification 20× and 40×, respectively.

**Figure 6 ijms-25-09711-f006:**
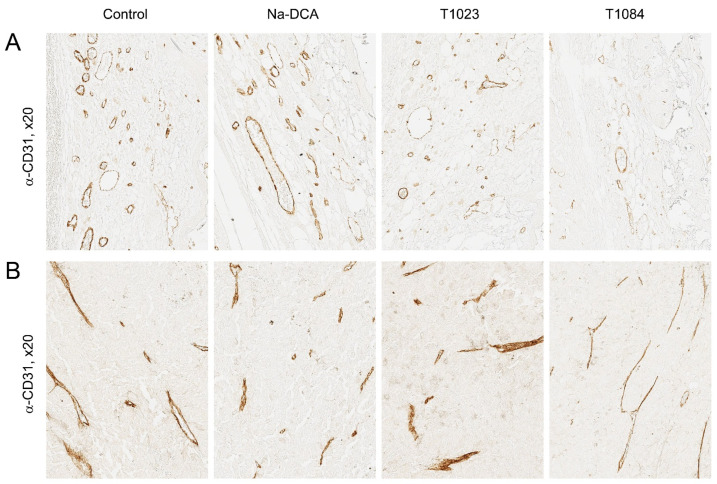
Effect of T1023, Na-DCA, and T1084 with chronic i.p. administration at equimolar doses 220 μM (60.0, 33.6, and 70.7 mg/kg, respectively) on the vessels of B-16 melanoma in female mice C57Bl/6j on the 6th day after the beginning of exposure. Immunostaining with antibodies to CD31, objective magnification 20×. (**A**) Peritumoral area of tumor nodes. (**B**) “Hot spots” of angiogenesis in the parenchyma of tumor nodes.

**Figure 7 ijms-25-09711-f007:**
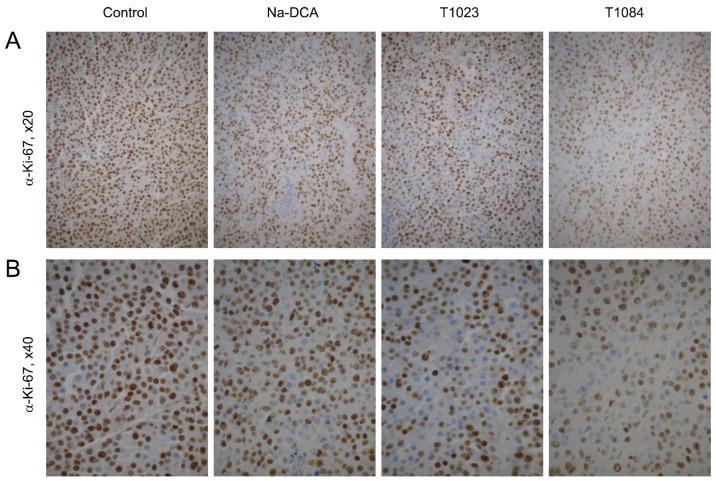
Effect of T1023, Na-DCA, and T1084 with chronic i.p. administration at equimolar doses 220 μM (60.0, 33.6, and 70.7 mg/kg, respectively) on the proliferative activity in the parenchyma of melanoma B-16 tumor nodes on the 6th day after the beginning of exposure. Immunostaining with antibodies to Ki-67, objective magnification ×20 (**A**) and ×40 (**B**).

**Figure 8 ijms-25-09711-f008:**
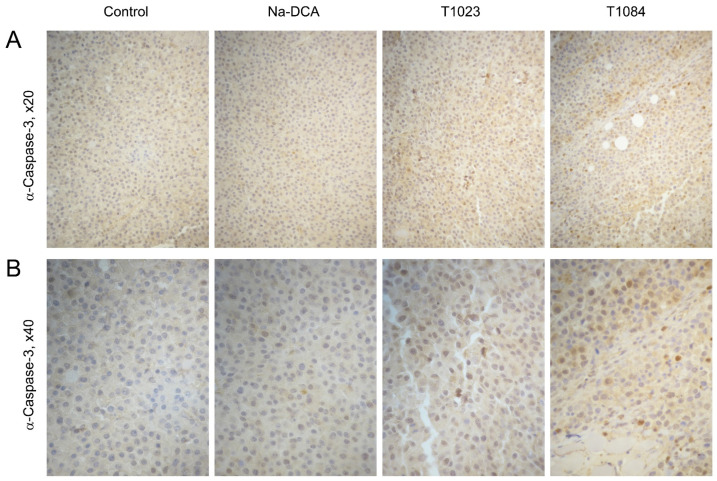
Effect of T1023, Na-DCA, and T1084 with chronic i.p. administration at equimolar doses 220 μM (60.0, 33.6, and 70.7 mg/kg, respectively) on the apoptotic cell death in the parenchyma of melanoma B-16 tumor nodes on the 6th day after the beginning of exposure. Immunostaining with antibodies to Caspase-3, objective magnification 20× (**A**) and 40× (**B**).

**Table 1 ijms-25-09711-t001:** General morphometric indicators of B-16 melanoma tumor nodes.

Group	Volume Content of Necrosis Area, %	Volume Content of Atypical Dystrophic Cells Area, %	Volume Content of Viable Parenchyma, %
Control	6.4 ± 3.7	6.9 ± 2.6	86.7 ± 3.5
Na-DCA	7.7 ± 3.4	9.3 ± 3.5	83.1 ± 3.7
T1023	7.0 ± 2.6	11.0 ± 2.6 α	82.1 ± 1.3 α
T1084	9.6 ± 5.6	21.2 ± 7.5 β γ δ	69.2 ± 12.3 β γ δ

Notes: Symbols—significant differences: α—in control mice vs. T1023-treated mice (*p* = 0.04473, *p* = 0.00912); β—in control mice vs. T1084-treated mice (*p* = 0.00247, *p* = 0.00247); γ—in Na-DCA-treated mice vs. T1084-treated mice (*p* = 0.00633, *p* = 0.00706); δ—in T1023-treated mice vs. T1084-treated mice (*p* = 0.01216, *p* = 0.00748).

**Table 2 ijms-25-09711-t002:** Indicators of angioarchitecture of B-16 melanoma tumor nodes.

Group	Volume Content of CD31-Positive Structures, %	Number of Vessels per 1 mm^2^	Average Cross-Sectional Area, µm^2^
Peritumoral areas of tumor nodes			
Control	8.2 ± 3.2	299.5 ± 29.8	193.2 ± 43.5
Na-DCA	6.9 ± 2.6	285.4 ± 40.7	184.3 ± 67.6
T1023	5.0 ± 1.8 α β	222.6 ± 33.4 α β	195.7 ± 51.4
T1084	4.3 ± 1.9 γ δ	187.7 ± 61.1 γ δ	186.3 ± 49.0
“Hot spots” of angiogenesis in the parenchyma of tumor nodes			
Control	5.4 ± 2.7	100.0 ± 20.5	97.1 ± 29.0
Na-DCA	4.5 ± 1.7	89.0 ± 24.8	109.1 ± 38.2
T1023	4.3 ± 2.2	105.2 ± 49.7	104.5 ± 47.8
T1084	3.9 ± 1.5	107.7 ± 43.1	96.7 ± 24.5

Notes: Symbols—significant differences: α—in control mice vs. T1023-treated mice (*p* = 0.01931, *p* = 0.00689); β—in Na-DCA-treated mice vs. T1023-treated mice (*p* = 0.03252, *p* = 0.00973); γ—in control mice vs. T1084-treated mice (*p* = 0.00415, *p* = 0.00208); δ—in Na-DCA-treated mice vs. T1084-treated mice (*p* = 0.01446, *p* = 0.00549).

**Table 3 ijms-25-09711-t003:** Indicators of proliferation in the parenchyma of B-16 melanoma tumor nodes.

Group	Proportion of Proliferating Cells, % (α-Ki-67 Staining)	Mitotic Index, % (H&E Staining)
Control	81.9 ± 4.5	0.94 ± 0.09
Na-DCA	73.6 ± 5.6 α	0.59 ± 0.12 α
T1023	74.2 ± 5.5 β	0.54 ± 0.07 β
T1084	67.4 ± 8.9 γ	0.44 ± 0.07 γ

Notes: Symbols—significant differences: α—in control mice vs. Na-DCA-treated mice (*p* = 0.00212, *p* = 0.04782); β—in control mice vs. T1023-treated mice (*p* = 0.00362, *p* = 0.04057); γ—in control mice vs. T1084-treated mice (*p* = 0.00017, *p* = 0.04057).

**Table 4 ijms-25-09711-t004:** Indicators of apoptotic death in the parenchyma of B-16 melanoma tumor nodes.

Group	Number of Apoptosis per 1 mm^2^ (α-Caspase-3 Staining)	Apoptotic Index, % (α-Caspase-3 Staining)
Control	11.9 ± 2.7	0.25 ± 0.06
Na-DCA	20.1 ± 11.0	0.42 ± 0.27
T1023	27.0 ± 11.6 α	0.59 ± 0.29 α
T1084	32.6 ± 17.7 β	0.63 ± 0.35 β

Notes: Symbols—significant differences: α—in control mice vs. T1023-treated mice (*p* = 0.01925, *p* = 0.01938); β—in control mice vs. T1084-treated mice (*p* = 0.00264, *p* = 0.00269).

## Data Availability

The data presented in this study are available at Appendix A.

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
