# Peer review of "In Vitro Cytotoxic Potential and In Vivo Antitumor Effects of NOS/PDK-Inhibitor T1084"

_ijms, 2024, doi:10.3390/ijms25179711_

Round 1

Reviewer 1 Report

Comments and Suggestions for Authors

The manuscript presents a novel and promising approach by combining NOS and PDK inhibitory functions within a single compound, T1084, which demonstrates superior antitumor activity compared to its precursor compounds. The study is well-executed, with a thorough experimental design that includes both in vitro and in vivo models, using multiple cancer cell lines and two different tumor models in mice. The detailed analysis, particularly the robust histological and immunohistochemical studies, provides valuable insights into the mechanisms underlying the observed effects. While the manuscript is generally well-written, the introduction and abstract could be more concise, and some methodological details could be moved to supplementary materials to streamline the main text. Despite these minor issues, the study significantly contributes to the development of multifunctional antitumor agents, and with some revisions, it is well-suited for publication. 

Comments on the Quality of English Language

Moderate, can be better! 

Author Response

Thank you very much for your attention and positive attitude to our work.

Thank you very much for your attention and positive attitude to our work.

We constructively perceived your comments and tried to take them into account when editing the manuscript.

Your remark:

”While the manuscript is generally well-written, the introduction and abstract could be more concise, and some methodological details could be moved to supplementary materials to streamline the main text.”

We did not change the introduction, as this sentence contradicts the opinion of another reviewer. However, we made the abstract more compact. We moved the details to supplementary materials.

Reviewer 2 Report

Comments and Suggestions for Authors

The manuscript by Marina Filimonova et al. entitled "In Vitro Cytotoxic Potential and In Vivo Antitumor Effects of NOS/PDK-Inhibitor T1084" aims to investigate the in vitro cytotoxic effects and in vivo antitumor effects of the NOS/PDK-Inhibitor T1084, comparing them with the parent compounds, T1023 and Na-DCA. The study uses morphological, histological, and immunohistochemical methods to evaluate the efficacy of T1084 in normal and malignant human cell lines, as well as solid tumor models in mice.

Points that need to be revised in the manuscript

- The description of the protocols needs to be more specific, including details on the preparation of the solutions, concentrations used, and storage conditions to ensure the replicability of the experiments. It would be important to create the experimental design for both the in vitro and in vivo studies, so that the reader will benefit from understanding the work.

- I believe that it is necessary to justify the reason for choosing the cell lines and animal models, explaining their relevance and representativeness for the types of cancer studied. This aspect could be discussed in the introduction.

- The cytotoxicity methodology needs to be more detailed, specifying the reagents used, incubation conditions and IC50 calculation method to ensure the accuracy of the results.

- In the in vivo study, the administration regimen needs to be detailed, including dose, route of administration and frequency, in addition to describing how tumor growth was monitored and measured during the studies.

- It would be interesting to discuss further in the manuscript the mechanisms of sensitivity and resistance to T1084, T1023, and Na-DCA. The molecular mechanisms that explain the variations observed in the response of the different cell lines and tumor models need to be discussed.

- How do the results obtained translate into potential clinical impact? What would be the next steps needed to validate the clinical use of T1084 in antitumor treatments? How do the antitumor effects of T1084 compare with anticancer treatments currently used in clinical practice?

- What factors can explain the greater sensitivity observed in certain cell lines (such as A375 and HeLa) to T1084 compared to others? Is there any specific characteristic of these cell lines that contributes to this difference?

- Observing the references used, I believe that more current literature needs to be used.

- The limitations of the study need to be included.

Comments on the Quality of English Language

 Minor editing of English language required.

Author Response

Thank you very much for your attention and positive attitude to our work.

We constructively perceived your comments and tried to take them into account when editing the manuscript.

Your remarks:

1 - The description of the protocols needs to be more specific, including details on the preparation of the solutions, concentrations used, and storage conditions to ensure the replicability of the experiments. It would be important to create the experimental design for both the in vitro and in vivo studies, so that the reader will benefit from understanding the work.

The methods for preparing all the solutions of the substances being studied, as well as the concentrations, volumes, methods, and schemes for their administration (or addition) in in vitro and in vivo studies are described in great detail in the Materials section. All solutions were prepared before use (which is also written). We did not study the issues of their storage conditions.

The aim of the work is to study the features of the NOS/PDK inhibitor T1084 antitumor activity. Therefore, the experimental design is obvious – a study of the cytotoxic and antitumor effects of T1084 in comparison with the effects of its precursor substances – T1023 (NOS inhibitor) and Na-DCA (PDK inhibitor), when they are used in equimolar doses (or concentrations). So in this case, this is an obvious, expedient approach that does not require justification or any separate description.

Moreover, your comment directly contradicts another reviewer.

At the same time, we moved section 4.3.1. (Compounds and Solutions Used in In Vivo Experiments) closer to section 4.3.4. (Treatment of Mice Tumors) so that they are not separated and the administration schemes in the continuation are clear. We have expanded section 4.2.3 and added a significant amount of data to the Supplements.

2 - I believe that it is necessary to justify the reason for choosing the cell lines and animal models, explaining their relevance and representativeness for the types of cancer studied. This aspect could be discussed in the introduction.

According to GLOBOCAN, breast cancer was one of the most frequently diagnosed cancers in 2022 (11.6%), followed by prostate cancer (7.3%), cervical cancer (3.3%), leukemia (2.4%), melanoma (1.7%) and ovarian cancer (1.6%) [Global cancer statistics 2022: GLOBOCAN estimates of incidence and mortality worldwide for 36 cancers in 185 countries. CA Cancer J Clin. 2024, 74, 229-263. doi: 10.3322/caac.21834]. In this regard, the choice of a panel of cell lines and animal tumor models was justified by both global statistics on cancer incidence and mortality (GLOBOCAN, https://gco.iarc.who.int/en ) and individual features of the metabolism of malignant neoplasm cells. Thus, according to the literature, ovarian cancer [Anwar, S., Shamsi, A., Mohammad, T., Islam, A., Hassan, M. Targeting pyruvate dehydrogenase kinase signaling in the development of effective cancer therapy. Biochim Biophys Acta Rev Cancer. 2021, 1876, 188568. doi: 10.1016/j.bbcan.2021.188568], breast cancer [Mishra, D., Patel, V., Banerjee, D. Nitric Oxide and S-Nitrosylation in Cancers: Emphasis on Breast Cancer. Breast Cancer (Auckl). 2020, 14, 1178223419882688. doi: 10.1177/1178223419882688] have a high level of PDK expression. And high expression of iNOS has been shown for prostate cancer [Liu, X., Zhang, Y., Wang, Y., Yang, M., Hong, F., Yang, S. Protein Phosphorylation in Cancer: Role of Nitric Oxide Signaling Pathway. Biomolecules. 2021, 11, 1009. doi: 10.3390/biom11071009.], cervical cancer [Function of inducible nitric oxide synthase in the regulation of cervical cancer cell proliferation and the expression of vascular endothelial growth factor. Mol Med Rep. 2014, 9, 583-589. doi: 10.3892/mmr.2013.1838]. However, the most favorable metabolic profile for the effect of the studied compound (T1084), characterized by increased expression of both iNOS and PDK, was found in melanoma [Populo, H., Caldas, R., Lopes, J., Pardal, J., Máximo, V., Soares, P. Overexpression of pyruvate dehydrogenase kinase supports dichloroacetate as a candidate for cutaneous melanoma therapy. Expert Opin Ther Targets. 2015, 19, 733-745. doi: 10.1517/14728222.2015.1045416; Nascimento, R., Nagamine M., Toledo, G., Chaible, L., Tedardi, M., Del-Grande, M., Fonseca, I., Dagli, M. Sodium dichloroacetate attenuates the growth of B16-F10 melanoma in vitro and in vivo: an opportunity for drug repurposing. Anticancer Drugs. 2021, 32, 111-116. doi: 10.1097/CAD.0000000000001013; Jimenez, J., Dubey, P., Carter, B. , Koomen, J., Markowitz, J. A metabolic perspective on nitric oxide function in melanoma. Biochim Biophys Acta Rev Cancer. 2024, 1879, 189038. doi: 10.1016/j.bbcan.2023.189038.]. This may have been the reason for its high sensitivity to the action of our multitarget compound.

The choice of in vivo tumor models was justified by the results obtained in in vitro studies – high sensitivity of the HeLa line (human cervical cancer cells) to the action of T1084 and high sensitivity of the A375 line (human melanoma cells) to the action of T1023 and T1084. The last paragraph of section 2.1 states this in plain text:

«These results justified the feasibility of studying the in vivo antitumor effects of T1084 in comparison with its precursor compounds, Na-DCA and T1023, on two solid tumor models in mice: cervical cancer CC-5 and B-16 melanoma».

In our opinion, such strains of animal tumors are quite adequate models of human cervical cancer and melanoma.

3 - The cytotoxicity methodology needs to be more detailed, specifying the reagents used, incubation conditions and IC50 calculation method to ensure the accuracy of the results.

The MTT test was proposed more than 40 years ago (Mosmann, T. Rapid colorimetric assay for cellular growth and survival: Application to proliferation and cytotoxicity assays. J. Immunol. Methods 1983, 65, 55–63). During this time, its methodological part and quantitative analysis have not undergone significant changes. Sections 4.2.1-4.2.3 describe in detail the composition of the medium, reagents, conditions and time of cultivation, concentrations of the substances being studied, and the method for measuring optical density using a vital dye. As described, cells were incubated in the presence of T1023 or T1084 at concentrations of 0, 62.5, 125, 250, 500 and 1000 µg/mL. The calculation of cell survival at the corresponding concentrations of T1023 and T1084 is described in section 4.2.3. The obtained concentration-survival data were analyzed using the logistic model of the Origin 6 software package (“Non-linear curve fit - Logistic”) and the concentration (in µg/mL units) at which half-maximal inhibition of cell growth was observed (IC50) was calculated. The obtained IC50 values for T1023 and T1084 were then converted to mM units by dividing by the corresponding molecular weight (this is described above in section 4.2.3). To enable verification of the accuracy of the MTT test results, a table (Table S.1) has been introduced in Supplements, reflecting the IC50 estimates for all cell lines obtained in independent experiments.

4 - In the in vivo study, the administration regimen needs to be detailed, including dose, route of administration and frequency, in addition to describing how tumor growth was monitored and measured during the studies.

Section 4.3.4 Treatment of Mice Tumors details the administration regimens, including dose, route of administration, and frequency. Section 4.3.5 Morphological Assessment of Effects describes how tumor growth was monitored and measured during the studies.

5 - It would be interesting to discuss further in the manuscript the mechanisms of sensitivity and resistance to T1084, T1023, and Na-DCA. The molecular mechanisms that explain the variations observed in the response of the different cell lines and tumor models need to be discussed.

The study of the mechanisms of sensitivity and resistance of neoplasms to therapeutic effects currently represents a huge area of knowledge, describing multiple factors that implement these phenomena.. In our limited experimental study, it is difficult to analyze this layer of knowledge.

6 - How do the results obtained translate into potential clinical impact? What would be the next steps needed to validate the clinical use of T1084 in antitumor treatments? How do the antitumor effects of T1084 compare with anticancer treatments currently used in clinical practice?

Based on the available data, it can be assumed that the area of ​​T1084 application is adjuvant therapy of solid malignant neoplasms characterized by a high level of vascularization. The results of the studies show that one of the proposed nosologies for the use of this agent could be melanoma.

To date, promising data have been obtained on the prospects of using T1084 in combination with chemotherapy (e.g., cyclophosphamide) or with radiotherapy (single and fractionated γ-irradiation). It has been shown that the course application of T1084 increases the antitumor effect of these agents without increasing the toxicity of the combination. In addition, according to modern literature, such agents may be effective in increasing the effectiveness of photodynamic therapy of metastatic aggressive neoplasms (which could be studied in the future).

The next steps to bring the T1084 compound closer to clinical use are the development of optimal regimens for its oral use. This is relevant because cancer patients often require long-term antitumor therapy, and this method of use will be the most optimal.

7 - What factors can explain the greater sensitivity observed in certain cell lines (such as A375 and HeLa) to T1084 compared to others? Is there any specific characteristic of these cell lines that contributes to this difference?

This also remains to be studied.

8 - Observing the references used, I believe that more current literature needs to be used.

Indeed, there are more recent studies. For example:

About antiangiogenic therapy and resistance to it:

  1. Eelen, G., Treps, L., Li, X., Carmeliet, P. Basic and Therapeutic Aspects of Angiogenesis Updated. Circ Res. 2020, 127, 310-329. DOI: 10.1161/CIRCRESAHA.120.316851

On the role of nitric oxide in angiogenesis:

  1. Miranda, K.M., Ridnour, L.A., McGinity, C.L., Bhattacharyya, D., Wink, D.A. Nitric Oxide and Cancer: When to Give and When to Take Away? Inorg Chem. 2021, 60, 15941-15947. doi: 10.1021/acs.inorgchem.1c02434.
  2. Alsharabasy, A. M., Glynn, S.A., Pandit, A. The role of extracellular matrix in tumour angiogenesis: the throne has NOx servants. Biochem Soc Trans. 2020, 48, :2539-2555. doi: 10.1042/BST20200208.

On the role of nitric oxide in melanoma progression:

  1. Jimenez, J., Dubey, P., Carter, B., Koomen, J., Markowitz, J. A metabolic perspective on nitric oxide function in melanoma. Biochim Biophys Acta Rev Cancer. 2024, 1879, 189038. doi: 10.1016/j.bbcan.2023.189038
  2. Ding Z., Ogata, D., Roszik, J., Qin, Y. , Kim, S., Tetzlaff, M.T., Lazar, A.J., Davies, M.A., Ekmekcioglu, S., Grimm, E.A. iNOS Associates With Poor Survival in Melanoma: A Role for Nitric Oxide in the PI3K-AKT Pathway Stimulation and PTEN S-Nitrosylation. Front Oncol. 2021, 11, 631766. doi: 10.3389/fonc.2021.631766. eCollection 2021.

However, the literature we used, although in some cases relatively old, but, in our opinion, is more appropriate in its detailed reflection or comprehensive analysis of the relevant phenomena.

9 - The limitations of the study need to be included.

Sorry, we didn't understand what was meant.

Round 2

Reviewer 2 Report

Comments and Suggestions for Authors

Most of the requested points have been addressed. However, one of the points that was requested but not addressed concerns the study's limitations. This refers to all the approaches that were not covered in the present manuscript, which serve as perspectives for other authors or researchers to explore in future works.

Comments on the Quality of English Language

Minor editing of English language required.

Author Response

Thank you very much for your kindness and sincere participation in improving our work.

We have taken careful note of your last comment as we understood it.

Your remark:

“Most of the requested points have been addressed. However, one of the points that was requested but not addressed concerns the study's limitations. This refers to all the approaches that were not covered in the present manuscript, which serve as perspectives for other authors or researchers to explore in future works.”

We believe that in the Results and Discussion sections we have sufficiently described and discussed the features of the antitumor effects of T1084.

In the last paragraph of the Discussion section we have indicated the main important issues that were not included in the scope of this work. The study of these issues will allow us to objectively assess the practical potential of this compound.

Your comments on the Quality of English Language:

“Minor editing of English language required.”

We have tried to carefully proofread the text and correct any errors.
